# Colloidal shuttles for programmable cargo transport

Ahmet F. Demirörs [ID] [1], Fritz Eichenseher[2], Martin J. Loessner[2] & André R. Studart [ID] [1]

The active transport of cargo molecules within cells is essential for life. Developing synthetic strategies for cargo control in living or inanimate thermal systems could lead to powerful tools to manipulate chemical gradients at the microscale and thus drive processes out of equilibrium to realize work. Here we demonstrate a colloidal analog of the complex biological shuttles responsible for molecular trafficking in cells. Our colloidal shuttles consist of magneto-dielectric particles that are loaded with cargo particles or living cells through size-selective dielectrophoretic trapping using electrical fields. The loaded colloidal shuttle can be transported with magnetic field gradients before cargo is released at the target location by switching off the electrical field. Such spatiotemporal control over the distribution of chemically active cargo in a reversible fashion can be potentially exploited for fundamental biological research or for the development of novel technologies for advanced cell culturing, drug discovery and medical diagnosis.

[1] Complex Materials, Department of Materials, ETH Zurich, 8093 Zurich, Switzerland. [2] Institute of Food, Nutrition and Health, ETH Zurich, Schmelzbergstrasse 7, 8092 Zurich, Switzerland. Correspondence and requests for materials should be addressed to A.F.Dör. (email: ahmet.demiroers@mat.ethz.ch) or to A.R.S. (email: andre.studart@mat.ethz.ch)

The transport of cargo is a widespread process in life[1, 2]. Depending on the length scale, cargo may refer to cars, people, goods, or molecules. In all these processes, cargo transport involves three basic steps: cargo loading onto a shuttle, directed motion of the cargo-shuttle assembly and cargo release to the destination. Biological shuttles are particularly impressive because they form useful chemical gradients in an otherwise thermally randomized environment[3, 4]. This is key to establish the non-equilibrium conditions that drive vital functions in living organisms[4]. Controlling cargo transport[5] and the spatial distribution of particles and molecules[6, 7] in thermal systems can be a powerful tool to manipulate the chemical environment in modern analytical technologies[8] for medical diagnosis[9] and drug discovery, such as organ-on-a-chip devices[10–13].

Here, we propose a colloidal analog of the carrier system used for the transport of biomolecules in living systems, which we name "colloidal shuttles". In contrast to the complex molecular machinery used in biology, our colloidal shuttles are driven and controlled purely by physical forces imposed by external magnetic and electrical fields. This is realized by designing a colloid shuttle in the form of particles that is simultaneously dielectric and superparamagnetic. Electrical fields are used to enable loading and release of cargo through modulation of dielectrophoretic forces in the vicinity of the colloidal shuttle. Directed motion of the cargo towards deliberate destinations is implemented by utilizing magnetophoretic forces imposed by a translating magnetic field gradient[14–17]. Since these fields are complementary to the types of chemical triggers exploited by biological systems and their artificial counterparts, our colloidal shuttles offer a powerful orthogonal strategy to design and control cargo transport in living and synthetic environments.

## Results

**Trapping and release mechanism.** We first describe the working principles underlying the electrically triggered cargo loading/unloading events using large and small silica particles as model colloidal shuttles and cargo, respectively. When exposed to an external electrical field, the dielectric colloidal shuttle modulates the electric field strength around itself to form a field gradient that works as a dielectrophoretic trap[18–23] for smaller cargo colloids. The dielectric trap is based on the inhomogeneous electric field generated by the shuttle particle, which essentially acts as a colloidal tweezer. A colloidal shuttle with a dielectric constant lower than the dielectric constant of the medium weakens the electric field strength at the poles of the colloid along the electric field direction, while increasing the field at the equator of the sphere. We quantify this effect by performing finite element calculations (Comsol Multiphysics 5.2). The electric field strength modulations caused by a rod-shaped tweezing particle with lower dielectric constant compared to the solvent placed between two electrodes are shown in Fig. 1. A continuous medium with a dielectric constant $\varepsilon_m = 47$ was chosen in this example, since this is the dielectric constant of the solvent dimethyl sulfoxide (DMSO) used in our experiments shown in Fig. 2. Surface plots of the electric field strength at different cross-sections of the sample cell are shown in Fig. 1a–c. Figure 1a displays the field strength in the $xy$ plane right above the rod, which clearly shows

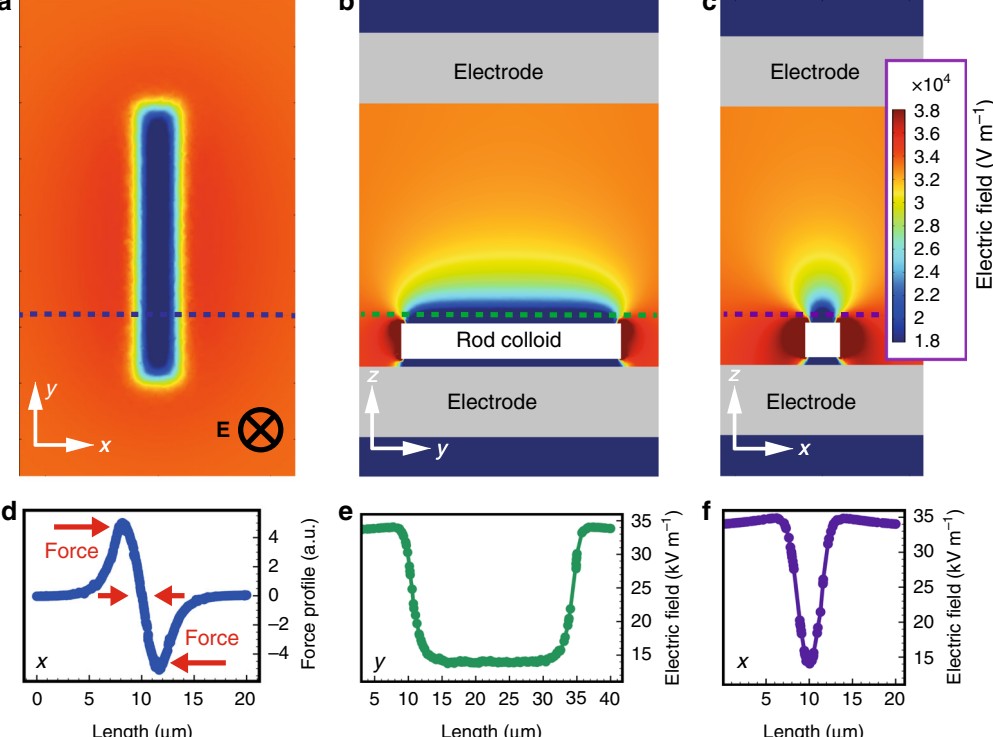

**Fig. 1** Finite element analysis of the electric field strength around a colloid. The tweezing colloidal particle in the vicinity of one of the two parallel electrodes changes the local electric field strength around itself. **a** Electric field strength right above the rod-shaped colloid is given in the $xy$ plane. Electric field strength surface at the cross-section of the colloid are given across the long (**b**) and short (**c**) axes. Corresponding line plots of the electric field strength along the long (green) and short (purple) axes are given in **e**, **f**, respectively. Smaller cargo particles with a negative dielectric constant contrast to the suspended medium DMSO will experience a dielectrophoretic force towards the center of the rod. **d** The force profile along the $x$-axis at the position indicated by the blue dashed line in **a** above the colloidal rod is calculated for smaller cargo particles with a negative dielectric contrast. Particles will be trapped at the intersection where the net force is zero

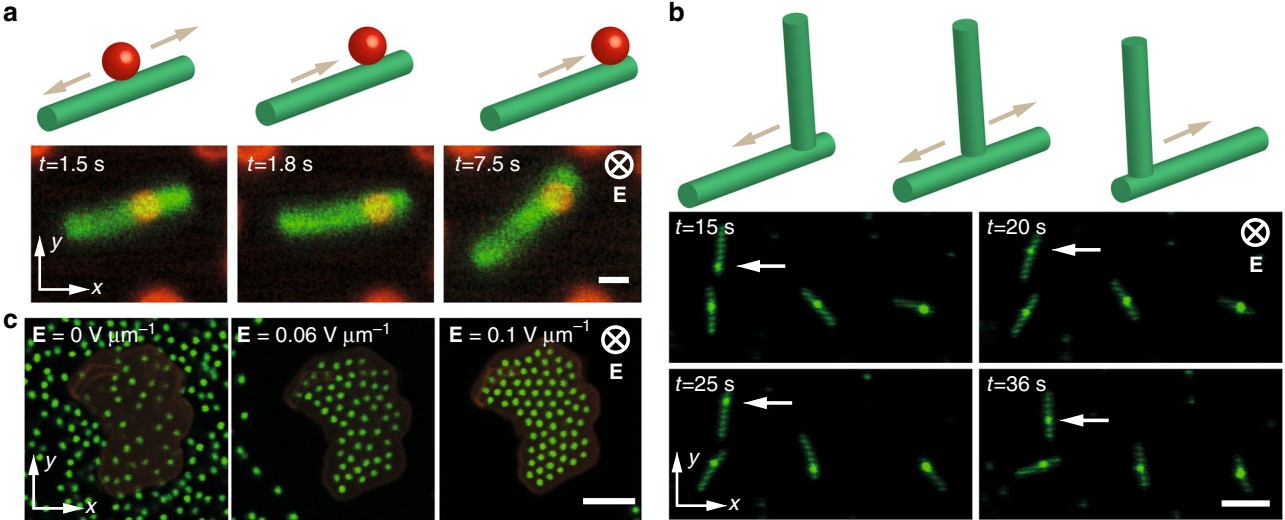

**Fig. 2** Trapping and assembly of cargo colloids by various colloidal shuttles. Rod-shaped particles act as colloidal shuttle, which trap other smaller colloids used as cargo colloids. Because the shape of the trap is anisotropic as the colloid, the smaller Brownian cargo explores the space above the colloidal shuttle along the long axis of the rod. **a** A 1 μm sized fluorescent silica colloid (red) is trapped over a larger silica rod (green) with an aspect ratio of 5. Above panels in **a**, **b** depict side view of the motion and assembly of colloids over time. **b** Horizontally lying colloidal rods trap others rods, which are aligned along the electric field. Standing rods move along the lying rod over time. **c** An alumina platelet labeled with rhodamine is used as a colloidal shuttle to trap 1 μm sized fluorescent silica colloids. In the absence of electric field particles are spread all over the space. When a field is applied particles start to move towards the alumina particle and accumulate over it. Further increasing the field increases the local concentration of the smaller colloid and result in a phase change from a liquid to a crystal. Scale bar in **a** is 1 μm and in **b**, **c** is 5 μm

how the field is reduced in the presence of the silica colloid of lower dielectric constant ($\varepsilon = 3.8$). Figure 1b, c depict the surface plots of the field strength at the cross-sections along the long ($yz$ plane) and short ($xz$ plane) axes of the colloidal rod, respectively. The dramatic drop of the field strength above the rod forms a potential trap for other smaller suspended colloids, as the electric potential energy scales with the square of the field: $U_E \sim |\mathbf{E}|^2$.

Small cargo particles approaching such an inhomogeneous electric field experience a force that depends on the dielectric constant contrast between the medium and the small particles. This so called dielectrophoretic (DEP) force experienced by the particles is given by the following formula[18, 19]:

$$\mathbf{F}_{DEP} = 2\pi a^3 \varepsilon_m \mathrm{Re}\left\{\frac{\varepsilon_m^* - \varepsilon_p^*}{\varepsilon_m^* + 2\varepsilon_p^*}\right\}\nabla|\mathbf{E}|^2 \qquad (1)$$

where $\varepsilon_{p,m}^*$ is the complex permittivity term, $\varepsilon_{p,m}^* = \varepsilon_0\varepsilon_{p,m} - i\sigma_{p,m}/\omega$; the subscripts $p$ and $m$ refer to the particle and medium; $\varepsilon$ and $\sigma$ refer to their relative permittivity and conductivity, respectively; $a$ is the radius of the particle, and $\mathbf{E}$ is the electrical field. The factor in brackets is known as the complex Clausius–Mossotti function[24] and contains the frequency dependence of the DEP force.

The dielectrophoretic force enables the colloidal shuttle to capture cargo particles. Such physical mechanism can be used to trap suspended colloids within the whole dielectric range. However, the position of the trapped particle around the colloidal shuttle will vary. For the colloidal shuttle with lower dielectric constant ($\varepsilon < \varepsilon_m$, negative polarization) used in our calculations and experiments, cargo particles with $\varepsilon_p < \varepsilon_m$ will be trapped at the top and bottom poles while particles with $\varepsilon_p > \varepsilon_m$ will be trapped at the equator, as shown in Supplementary Figs. 1, 2. Reversed configurations should form if a colloidal shuttle with dielectric constant higher than that of the medium is used ($\varepsilon > \varepsilon_m$). Irrespective of the polarization of the shuttle and the cargo, there will always be an attraction between them provided that

they are polarized with respect to the continuous medium in which they are suspended. However, the position of attraction will vary between the poles and the equator, depending on the combination of polarizations of cargos and shuttles. Because the electric field gradient and the DEP forces around the colloidal shuttle stays essentially constant as a function of its elevation from the bottom electrode, we are confident that this technique operates in 3D with the same efficiency as on the surface of the electrode (Supplementary Fig. 3).

Figure 1d shows the calculated force profile (**F**) experienced by the small cargo particles along the x-axis (blue dashed line in Fig. 1a) above the rod-shaped colloidal shuttle. According to this force profile, particles will be trapped at the center of the rod where the force is equal to zero. The slope of the force-distance curve around this equilibrium point indicates that when the particle moves away from this spot, it experiences a positive or negative force opposing its motion. One can assume that such a trap behaves like a spring and in analogy to the spring constant from Hooke's law, the negative of the first derivative of force will yield the stiffness[25] of the trap, $k = -\frac{\partial \mathbf{F}}{\partial x}$ (Supplementary Fig. 1). Because of its anisotropic shape, a rod-shaped particle creates traps with different dimensions along the distinct axes of the colloidal shuttle. Figure 1e is the plot of the electric field strength above the colloidal rod along the long axis (green dashed line, y-axis), as shown in Fig. 1b. Figure 1f is the plot of the electric field strength right above the colloidal rod along the short axis (purple dashed line, x-axis) as shown in Fig. 1c. As a result of the anisotropic nature of this particular trap, cargo colloids are expected to be fixed along the transverse axis (x) but free to move laterally along the longitudinal axis (y).

To demonstrate the trapping action predicted by the simulations, we performed an experiment with 4.6 μm long rod-shaped colloids as shuttles and 1 μm sized spherical silica colloids as the cargo (Fig. 2a). The cargo colloid is fluorescently labeled red, whereas the colloidal shuttle lying on the electrode is labeled green. Due to its small size and Brownian nature the colloidal

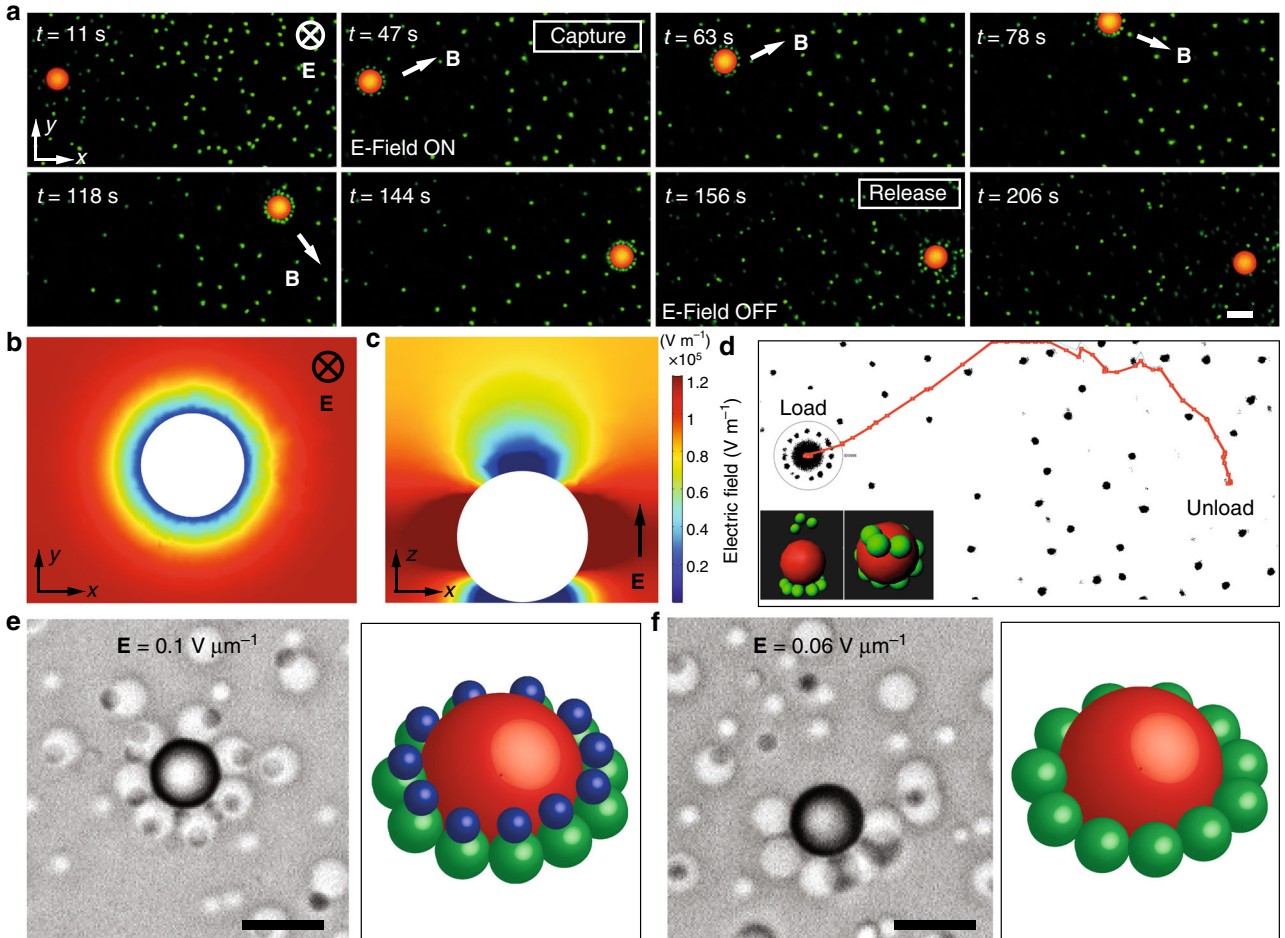

**Fig. 3** Cargo transport and delivery with size-selectivity. Superparamagnetic polymer colloids are used as carrier-colloids to pick up, transport and unload the colloidal cargo. **a** An external magnetic field gradient **B** is used to relocate the particles whereas electric field is used to pick up and release the cargo colloids. Snapshots show how the initially free particles are trapped around the larger superparamagnetic (red) particle using an electric field. Applying magnetic field to this particle assembly enables transport of the cargo (green particles) to the destination. When the electric field is turned off, trapped particles are released to the medium. **b**, **c** Calculations of the electric field strength around such colloidal shuttle (red in **a**) in $xy$ plane (**b**) and $xz$ plane (**c**). Field strength values are shown in V m$^{-1}$. Both calculations show that colloids will be trapped around the carrier-colloid in a ring fashion slightly below the equator of the carrier-sphere and on top of the carrier at the upper pole. **d** Path that the shuttle-cargo assembly takes throughout the time series of images given in **a**. Insets show 3D reconstructions of the particle assembly with different perspectives, namely side view and top view. **e**, **f** A colloidal shuttle can selectively trap large (2 μm silica) and/or small (1.5 μm silica) cargo depending on the applied electric field. Both particle types are trapped at **E** = 0.1 V μm$^{-1}$ field strength (**e**), whereas only large particles are trapped at a lower **E** value of 0.06 V μm$^{-1}$ (**f**). Scale bars are 4 μm

shuttle dynamically moves on the surface of the electrode. Indeed, the rod acts as a line trap for the spherical particle, as predicted by the calculations shown in Fig. 1. This causes the cargo colloid to remain confined transversely while freely floating above the surface of the shuttle along the longitudinal direction.

The trapping effect of the shuttle is not limited to spherical cargos. If the silica rods themselves are used as cargo, they remain effectively trapped on top of the rod-shaped shuttle (Fig. 2b). In this case, silica rods polarize with the field and thus align perpendicular to the surface of the electrodes, which is energetically favorable[26]. However, some silica rods that initially sedimented on the electrode cannot stand up due to the image charges that attract them to the electrode surface[27]. Images in Fig. 2b are snapshots that demonstrate the motion of trapped standing rods along the lying ones. Note that the standing particles appear like spherical particles as they are scanned across their short axes. White arrows highlight one of the rods that moves up and down along the axis of the lying rod over time. Sketches above the images in Fig. 2a, b show a perspective of how trapping and motion takes place (Supplementary Movies 1, 2).

Making the colloidal shuttle large enough relative to the cargo colloids allows the Brownian trapped particles to explore different two-dimensional configurations on the surface of the shuttle. As an example, Fig. 2c shows a large-area colloidal shuttle consisting of a non-Brownian alumina platelet, 8.3 μm-long and 400-nm-thick, covered by a thin red-fluorescent coating[28]. Here, the alumina shuttle is shown to trap 1 μm sized green-fluorescent silica colloids. In the absence of an electric field, particles are spread all over the place. When a field of 0.06 V μm$^{-1}$ is applied, particles start to move and accumulate over the platelet, forming a disordered local assembly. Remarkably, further increasing the field to 0.1 V μm$^{-1}$ increases the local concentration of the colloids on the platelet until a phase transition into a 2D crystalline arrangement of cargo particles occurs[21, 29]. The ability to reversibly induce such order-disorder phase transitions allows us to program the assembly of cargo on the surface of the shuttles by simply modulating the applied electrical field.

**Programmed cargo transport.** Colloidal shuttles carrying trapped cargo colloids can be transported using an external magnetic

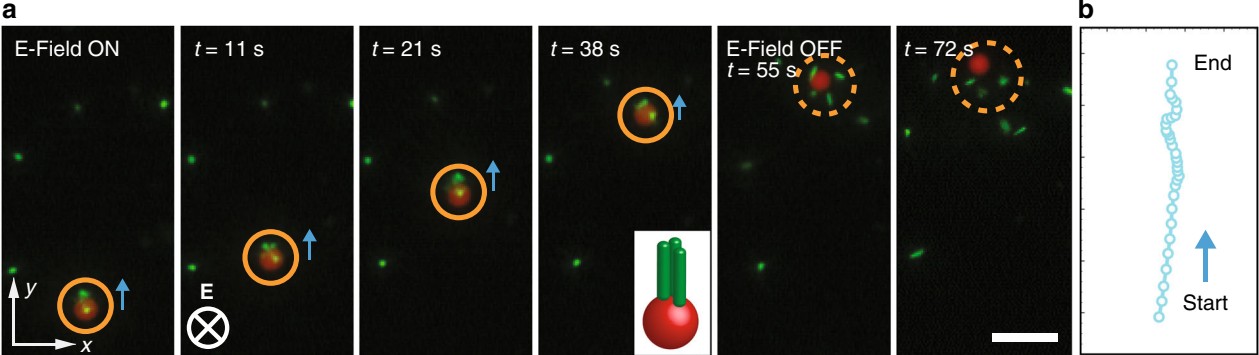

**Fig. 4** Active transport of *E. coli* bacteria by a superparamagnetic colloidal shuttle. A superparamagnetic polystyrene particle is used as a colloidal shuttle to transport *E. coli* as a model bio-cargo. **a** Snapshots showing the motion of the shuttle and the on-demand release of bacteria. **b** The trajectory of the colloidal shuttle is drawn based on the time series shown in **a**. Scale bar is 20 μm

field if a magnetic mismatch is introduced in the system. Provided that it has the necessary dielectric properties, superparamagnetic colloidal shuttles can be used to generate such magnetic mismatch relative to the continuous phase. Here, we use spherical polystyrene particles loaded with iron oxide (~20 w%) as an example of a superparamagnetic colloidal shuttle (Methods section). Such composite particle shows a dielectric constant of about 4.5, comparable to the previous experiments with silica. When suspended in a solution like DMSO ($\varepsilon = 47$) or mixtures of DMSO and water ($\varepsilon = 80.4$) or in deionized water (Supplementary Fig. 4), these superparamagnetic particles will have sufficient dielectric permeability contrast with the medium to generate dielectrophoretic traps while also responding to magnetic field gradients. Thus, such magneto-dielectric colloid was combined with spherical silica cargo to generate responsive shuttle-cargo clusters.

Exposing the resulting shuttle-cargo colloidal cluster to electrical and magnetic fields enables pick-up, transfer and release of a wide variety of objects, including inanimate colloids and living cells. Figure 3a shows an example of a delivery experiment demonstrated by using a red-fluorescent superparamagnetic particle as colloidal shuttle (4.69 μm). Panels of Fig. 3a depict the position of the colloidal shuttle and the cargo over time. Green-fluorescent silica colloids with size of 1 μm are used as model cargo. Free silica particles reversibly attach to the colloidal shuttle when the electric field is turned on at $t = 47$ s. By using a thin rod-shaped permanent magnet, we transport the superparamagnetic colloidal shuttle together with its cargo over a distance of several tens of microns. This was accomplished by rotating the permanent magnet ~2 cm away from the microscope objective. In response to this rotating field the cargo-shuttle clusters roll on the surface of the electrode. Using this rolling mechanism for transport, the direction of the cargo-shuttle motion can be easily reversed by simply changing the direction of magnet rotation (Supplementary Figs. 5, 6, Supplementary Movies 6, 7).

Magnetic-assisted motion is shown by the snapshots starting at $t = 47$ until $t = 144$ s in Fig. 3a and by the cluster trajectory depicted in Fig. 3d. Release of the cargo takes place when the field is turned off at an elapsed time, $t = 156$ s. Shortly after the field is switched off at $t = 206$ s, all the cargo colloids spread away due to thermal randomization in the form of Brownian motion (Supplementary Movie 3). Figure 3b, c shows electric field strength calculations for such a colloidal shuttle in *xy* and *xz* planes, respectively. According to these calculations, trapping of the particles takes place in the lower ring below the equator of the colloid where the field gradient is highest. For the color coding

used in Fig. 3c, this corresponds to the position where the color changes from red to blue. There is also a trap on top of the colloidal shuttle, where we also observed colloids in the three dimensional volume scans made by confocal microscopy. This is shown from different perspectives in the inset of Fig. 3d.

**Transport of biologically relevant cargos and size-selectivity.** Because dielectrophoretic forces will apply for any volume with a dielectric contrast to the medium, our colloidal shuttle can also trap and carry bio-objects like living cells or bacteria suspended in water. Water in such a scenario will provide the necessary environment for living objects to survive but also the necessary dielectric contrast for dielectrophoretic trapping. To demonstrate this possibility and the versatility of the proposed shuttle system, we perform trapping and transferring experiments using *Escherichia coli* bacteria labeled with mTurquoise2 fluorescent protein as a cargo in water medium (Fig. 4). Due to their rod shape, the *E. coli* bacteria align along the field direction when an external field of 0.05 V μm⁻¹ is applied. Besides alignment, this field causes the *E. coli* to also gather on top of the magneto-dielectric colloidal shuttle. Next, we examine the possibility to carry and deliver the bacteria to another site on the substrate by applying a translating magnetic field gradient. Figure 4a depicts snapshots of the motion of the cargo-shuttle cluster rolling upwards with *E. coli* bacteria on its head, while the electric field was on. Turning the field off after 54.7 s led to the release of the *E. coli* bacteria. The trajectory of the colloidal shuttle is shown in Fig. 4b.

Reversible and multiple cargo loading and unloading events are also an important feature of biological shuttles[4]. The reversible nature of the pick-up and release events carried out by the colloidal shuttle was demonstrated by magnetically moving the free colloidal shuttle backward after unloading of the bacteria. When the shuttle was in the vicinity of the bacteria that it released on the way, we turned the field on again to re-capture the bacteria. The reversible pick-up of the cargo is shown in Supplementary Fig. 7, Supplementary Movie 4.

In addition to reversibility, colloidal shuttles are also able to size select the cargo to be transported, which to some extent resembles the selectivity of biological shuttles. The size selectivity of our colloidal transport system arises from the strong dependence of the dielectrophoretic force on the particle size ($\mathbf{F}_{DEP} \sim a^3$). This makes the shuttle more effective in trapping larger colloidal cargo. We illustrate this unique feature by exposing a colloidal shuttle to cargos of different sizes and monitoring the system while the external electrical field is increased. The results show that the larger colloidal cargo is

preferentially trapped on the surface of the shuttle at low applied electrical fields (Fig. 3e, f, see also Supplementary Movie 5). Here, fine-tuning of the applied electric field strength enables the selective capture of either the large particles at lower field strengths (E = 0.06 V µm$^{-1}$) or both types of particles simultaneously at higher field strengths (E = 0.1 V µm$^{-1}$). Note here that if both types of particles are trapped, the larger cargo particles will be captured first, forming a ring at the bottom of the shuttle. The polarization of the larger cargos that are first captured by the shuttle will attract and facilitate the subsequent pick-up of the smaller cargos. Note also that at high field strengths some dipolar cargo–cargo assemblies are observed (Supplementary Movie 5). While the interactions between the large and small cargo particles are attractive, the shuttle and the smaller cargo may interact repulsively, depending on the angle between them[30] (Supplementary Fig. 8). In addition to size selectivity, bio-specificity and chemical selectivity may also be achieved by using colloidal cargo functionalized with, for instance, streptavidin and biotin molecules.

To successfully carry the cargo colloids to specific destinations in these proof-of-concept experiments, we made sure that the drag forces exerted on the cargo was lower than the DEP forces that traps the cargo around the colloidal shuttle ($\mathbf{F}_{drag} \leq \mathbf{F}_{DEP}$, Supplementary Fig. 9). This was enforced either by applying a high enough electric field to keep the cargo trapped or by maintaining the speed of the cargo-shuttle cluster below a certain threshold. Taking electric field strengths values between 0.01 and 0.2 V µm$^{-1}$, we estimate from field strength gradients obtained via FEM analysis that the dielectrophoretic force ($\mathbf{F}_{DEP}$, Eq. (1)) should vary between 0.2 and 1.6 pN in the vicinity of the colloidal shuttle surface. This was compared with the drag force calculated using the Stoke's relation: $\mathbf{F}_{drag} = -6\pi\eta a\nu$, where η is the viscosity of the medium, $a$ is the radius of the particle and $\nu$ is the speed of the shuttle. Using the experimentally measured maximal velocities of the colloidal shuttles (2 to 9 µm s$^{-1}$) at several applied field strengths, the viscosity of DMSO ($\eta_{DMSO} = 1.99 \times 10^{-3}$ Pa s$^{-1}$), and a diameter of 1 µm for the silica cargo, we estimate the drag forces to be between 0.03 and 0.17 pN (Supplementary Fig. 9d). The lower drag forces compared to the estimated dielectrophoretic forces confirms that the chosen experimental conditions were appropriate for the successful entrapment of the cargo colloids during motion of the shuttle. Interestingly, we observe that when $\mathbf{F}_{DEP}$ is comparable to $\mathbf{F}_{drag}$ the cargo is only loosely trapped by the shuttle, which allows for shear-triggered cargo release[31] and for the dynamic exchange with other cargo particles present in the medium (Supplementary Fig. 9e). The triggering shear forces can be controlled by either changing the speed of the colloidal shuttle or flowing a fluid around a shuttle that is locked in place using an external magnetic field. In addition to programmable cargo release, our colloidal shuttle system can thus also be used to experimentally measure dielectrophoretic forces between colloidal particles to assess the validity of theoretical predictions[30].

Using this force balance, we can also predict the limits under which the dielectrophoretic trapping mechanism will work. Since the drag force scales linearly with the size $a$ ($\mathbf{F}_{drag} \sim a$) and the DEP force increases proportionally to the volume ($\mathbf{F}_{DEP} \sim a^3$), drag forces should not be able to displace larger cargo colloids from the shuttle. Therefore, the upper limit size is simply determined by the length scale of the dielectrophoretic trap. By contrast, smaller cargo particles can be removed from the shuttle by drag forces exerted by the fluid during motion. This lower limit can be estimated by balancing the DEP and drag forces, which leads to the following prediction: $a_{min} = \mathbf{F}_{DEP}/6\pi\eta\nu$. The magnitude of the DEP force can be calculated using Eq. (1) with the help of finite element modeling, as explained above.

Assuming a DEP force of 0.15 pN, we estimate for example a lower particle size limit of 200 nm[32] for the motion velocities and medium viscosity used in this work. Smaller cargo can also be trapped as long as the velocity of the shuttle or the viscosity of the fluid is further reduced. These predictions assume that the DEP forces are sufficiently strong to retain the cargo attached even at static conditions. To demonstrate the validity of this assumption we compare the trapping energy associated with the DEP force with randomizing thermal energy. Using silica colloids with a diameter of 5 µm, for example, DEP forces up to ~10 pN and electric potential energies on the order of ~10$^4$ $k$T are expected under the conditions utilized in our experiments.

In summary, the colloidal shuttles proposed offer a powerful tool to manipulate the distribution of inanimate or living species away from equilibrium at the microscale, analogous to the action of biological shuttles within cells. Given its purely physical nature, our approach is compatible to other possible triggers like chemicals, temperature or light, which could add further functionalities and controllability to colloidal shuttles. Established chemical strategies for particle surface functionalization could be exploited, for example, to anchor drug molecules on the surface of the cargo colloids or to program this surface to achieve chemical selectivity in the uptake and release processes. The controlled manipulation of these molecules can potentially be used to create microscale chemical gradients or to provide spatiotemporal control over the release of minor quantities of potential therapeutics or biomolecules in cell cultures[8] and bio-analytical platforms[9]. Chemical gradients can be created by locally releasing high concentrations of chemical species that are previously incorporated in porous or hydrogel cargo particles. The controlled release of these species from the cargo should be possible by following established strategies involving for example light or local heating as triggers. Utilizing such colloidal shuttles to controllably interact with tissues and organs recreated on a chip is another enticing possible application of this new technology.

## Methods

**Colloids and E. coli.** The spherical core-shell silica particles used as colloidal cargo had a fluorescent core of diameter $d \approx 500$ nm, surrounded by a non-fluorescent shell with a thickness of 250 or 500 nm. Such particles were synthesized using a protocol proposed by van Blaaderen et al.[33]. Suspensions were prepared by mixing these cargo particles with different types of colloidal shuttles. The following colloidal shuttles were tested: (i) rod-shaped silica particles with a length of 4.65 ± 0.4 µm and width of 752 ± 117 nm synthesized according to Kuijk et al.[34]; (ii) alumina platelets, 8.3 µm-long and 400-nm-thick purchased from Merck KGaA, Germany (White Sapphire grade) coated with a thin layer of silica for fluorescent labeling[28]; and (iii) superparamagnetic colloids 4.69 µm in size purchased from Microparticles GmbH (PS-MAG-RhB-S2538). For shuttling experiments, a variety of solutions could be used and all provided the necessary tools for our trapping approach to function. Initial experiments with particle mixtures were dispersed in a mixture of water (11.6 wt%) and dimethylsulfoxide (DMSO, 88.4 wt%) to match the refractive index of silica (1.45) or in pure DMSO. However, use of DMSO is not obligatory for such experiments as water will also exhibit similar behavior. Experiments with E. coli were performed in deionized water. E. coli bacteria was labeled with mTurquoise2 fluorescent protein.

**Electrode and sample fabrication.** Transparent and conductive indium tin oxide (ITO) cover slips were used as electrodes for applying the AC electric field at 1 MHz frequency while keeping the conductive faces parallel to one another. The sample cells were fabricated by sandwiching two ITO cover slips separated by spacers with thicknesses of 0.09–0.12 mm (#0 cover slips) or 0.06–0.08 mm (#00 cover slips).

**Finite element analyzes.** Finite element analyzes were conducted by using Comsol 5.2 to estimate the electric field strength landscape modulated by the shuttle particles. Such simulations are semi-quantitative and do not include the dielectric constant, the electrical conductivity and the frequency dependence of the dielectric response of the cargo colloid. These calculations provide useful information on how the field strength changes in 3D for a given potential applied between the electrode (0.1 V µm$^{-1}$ in our case). Taking cargo particles that display a negative or

positive mismatch relative to the dispersing medium, such analyzes help to predict several features of the cargo-shuttle assembly (see main text).

**Imaging and microscopy**. Our imaging and shuttling experiments were performed using a Leica SP2 inverted confocal scanning laser microscope with a PL APO ×63/1.40 NA objective. For *E. coli* and magnetic rolling experiments we used a fluorescence microscope with an inverted DM6000 stage and a mercury lamp (Leica).

**Magnetic gradient manipulation and rolling of the superparamagnetic colloids**. For magnetic gradient manipulation of the superparamagnetic colloid we used a thin neodymium magnet. Such magnet was 2 cm in diameter and generated a field of 0.2 T on the surface. As the magnet was brought close to the sample we observed the colloid cluster to move as a response to the field gradient generated by the magnet. For rolling the superparamagnetic colloids on the ITO glass slide, we rotated the magnet with a frequency of 1 Hz at a distance of 2 cm from the sample cell. The direction of magnet rotation dictated the rolling direction of the superparamagnetic colloidal shuttle.

**Data availability**. The data that support the findings of this study are available from the corresponding authors on request.

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

## Acknowledgements

We thank ETH Zürich, the Swiss National Science Foundation (Ambizione grant, number PZ00P2_148040) and Swiss National Center of Competence in Research (NCCR) for Bio-Inspired Materials for financial support. We also thank ScopeM, the microscopy center and FIRST, the cleanroom facility at ETH.

## Author contributions

A.F.D. designed and carried out experiments and calculations, F.E. and M.J.L. provided the biological samples and contributed to those experiments; A.F.D. and A.R.S. conceived the project and wrote the paper. All authors contributed to the writing of the manuscript.

## Additional information

**Competing interests:** The authors declare no competing financial interests.

