## [Peer Review File · Nature Communications]

Reviewers' comments:

Reviewer #1 (Remarks to the Author):

Demirors et al. have placed dielectric microparticles into DMSO (and a water/DMSO mixture consisting of 88% DMSO), applied an AC electric field ($f = 1$ MHz) and observed that these microparticles (termed shuttles) attract other microparticles (referred to as cargo) through electrostatic interactions. By switching the electric field off, the interaction could be turned off. The authors proceed to show that if superparamagnetic particles are used as shuttles, magnetic forces can be used to manipulate shuttles. The authors demonstrate that *E. coli* bacteria can be captured and manipulated by the shuttle particles. Comsol simulations are used to model the observations qualitatively.

The authors posit that the system is an analog of intracellular cargo transport.

Overall, the work is cute but I fail to see the major advance.

Apparently the system does not work in pure water or buffer, which makes it almost useless for biological systems. The creation of field gradients of ~ 10 V/mm is difficult outside of the flat cells used by the authors. The interaction between shuttle and cargo is non-specific in a biological sense, the shuttle simply collects any microscale particle in its environment.

The principles involved in the electrostatic interaction have been worked out in detail previously by Bevan et al.

The controllability of the shuttle motion is good in free space, but an actual target will in itself affect the electric field distribution and interfere with loading/unloading.

Finally, the system is lacking major aspects of biological active transport, such as its selectivity and autonomy. These aspects have been reproduced already in other colloidal systems.

The authors frequent invocation of chemical gradients seems inappropriate, since the transport of a few colloidal particles does not meaningfully change chemical potentials. Even for biological active transport, describing it in chemical terms is largely unhelpful.

Minor:

- 0.7 piconewtons should be abbreviated as 0.7 pN not 0.7 pNs.
- Fig. S1: What are the y-axis labels and units on the left and middle figure.
- Fig. S2a, 1c: What are the units of the color scale?
- What is the particle size when calculating $F(\text{DEP})$?

Reviewer #2 (Remarks to the Author):

A key factor in enabling construction, control, and characterization of systems in the nano and microscales is the ability to perform spatial manipulation of objects in these length scales. In recent years, nano and micro-shuttles, swimmers and rockets have been developed and introduced new capabilities, including controlled directional movement and transport of molecular and nanoscale cargo.

One of the major challenges in utilizing such systems for transport is obtaining controlled

mechanism for pick up and release of cargo. In this manuscript, Demirörs et. al present the use of dielectrophoresis (DEP) for controlled loading and unloading of microscale cargo onto colloidal magnetic shuttles, and the ability to load the cargo, transport the shuttle, and then unload it by combining external electric and magnetic fields.

In order to load cargo particles on the shuttle the authors utilize DEP forces between the particles that are formed when the particles are subjected to a (uniform) extremal electric field. The authors demonstrate using a finite element model, that when a colloidal particle (the shuttle) is immersed in a solvent with higher dielectric constant and subjected to a (uniform) extremal electric field, it perturbs the electric field in its vicinity. The electric field around the shuttle is reduced in the direction of the external field, and increased in perpendicular to it. The perturbation leads to a large gradient in the field around the particle, and consequently to a net DEP force, which can trap other (smaller) colloidal particles around the shuttle particle. Furthermore, the field depends on the geometry of the particles, and in anisotropic particles on the position of the particles with respect to the field.

The authors exhibit the applicability of DEP for cargo trapping using several systems of different geometries. Using time-lapse micrographs, the authors show that elongated rod-like shuttles act as 1d traps, which allows smaller spherical colloids or other elongated rods to move along the elongated shuttle. For large flat shuttles, the authors demonstrate accumulation of large number of spherical dots upon applying voltage, and a reversible transition of the spherical dots from disordered (liquid) to ordered structure (2d crystal) when the voltage is further increased. The authors also show that tuning the applied voltage can be used to form size-selective attachment of cargo particles on to the shuttle.

Combining the DEP loading-unloading mechanism with superparamagnetic shuttles allows controlled transport of the cargo against the concentration gradient. The authors demonstrate that using iron-oxide loaded polystyrene (with diameter of 4.6 micron) as shuttles allows transport of cargo (1 micron particles) to distances of tens of microns with velocities of ~ 1 micron per second. The authors demonstrate transport of inanimate cargo, as well as transport of non-spherical and cargo, including E-coli bacteria.

The method presented by the author offers an impressive demonstration of the potential of DEP forces for exciting applications including self-assembly, constraint diffusion, and controlled transport in the microscale under non-equilibrium conditions using external fields. The authors show the generality of the method by examining several systems of different geometries, demonstrate interesting features including 1d traps and liquid-to-crystalline transitions, and analyze the boundaries of the transport with respect to drag forces.

However, while the experimental results are indeed impressive, the treatment of the forces analysis, both theoretically and the experimentally should be further developed in order to better understand the behavior and the characteristics of the method. There are several issues which should be elaborated in the paper:

1. The major issue that is missing is a more comprehensive theoretical treatment of the forces. In particular, in all the calculations the authors model the electric field around the shuttle that results from the external electric field. However, they completely neglect (and ignore) the electric field around of cargo and the forces it exerts on the shuttle. Because the difference in size of the two is usually not very large, one might expect that both the cargo and the shuttle will have a mutual DEP force, which is not negligible. While exact treatment of the dynamics is complex as it includes both hydrodynamic and electrodynamic terms, there are several works that tackled the problem (for example Ai and Qian, *Journal of Colloid and Interface Science* 346, 2010, Pages 448-454), and might give better description

- of the systems). Such papers were not mentioned in the references, and I urge the authors to look deeper into inter-particle DEP interactions and incorporate it in this paper.
2. In the discussion, the authors mention that cargo attached using DEP might detach upon transport because of the drag force, and predict the maximum velocity that can be obtained before detachment. While in the scope of the paper this is an unwanted result, the detachment velocity can act as direct measurement for obtaining the PED forces experimentally. In addition to the information that such experimental measurement will contribute to this paper, it will be an excellent tool for assessing theoretical results regarding the inter-particle PED, which is still a subject of research. Thus, I would recommend measuring the detachment as function of velocity, and extracting an estimate for the forces involved in the PED shuttle-cargo systems.
 3. In all the calculations, the shuttle particle is on top of the electrode. What is the dependence of the shuttle field on the distance from the electrode? Can this approach be extended to 3D systems as well (for example transport in tissues or hydrogels)?
 4. Figure 3, panels e and f – please provide a more detailed information regarding the image. In particular, why would the small particles (depicted in blue) be positioned on top of the green particles? Why do the small particles seem darker in the micrograph images?

Thus, I recommend this paper for publication after major corrections, I particular regarding the theoretical calculation and experimental measurements of the forces involved in the process.

Reviewer #3 (Remarks to the Author):

This paper describes a clever way to collect, transport, and deliver colloidal cargos by combining dielectrophoresis and magnetic manipulation of superparamagnetic particles. The authors demonstrate reversible capture and release of colloidal cargo by using dielectrophoretic force and directed translational motion of colloidal shuttle using magnetic field gradient. Finite element calculations are performed to understand electric field strength around the colloidal shuttle. Both dielectrophoretic and magnetic manipulation of colloids are both well-known, the combination of the two into a single system is a fascinating study of controlled transport of micro cargos. It's not clear if this occurs only in DMSO/water or if it is also amenable to more biologically-relevant environments, but the manuscript is well-organized, well-presented, and should be of broad interest. After careful reading, there are a few issues that need clarification or correction:

1. The manuscript demonstrates the dielectrophoretic assembly of smaller particles onto larger ones, but there isn't any obvious reason why smaller particles shouldn't assemble with each other. Indeed, this seems to occur for rod-shaped colloids (Figure 2b), but is not observed between silica colloids, except perhaps in the colloidal crystal phase at higher field strengths (Figure 2c). From eq 1, it could be argued that the r^3 dependence would substantially reduce the interaction between smaller particles, but does not rule it out completely. The authors may want to clarify this issue.
2. The use of DMSO reduces the utility and applicability of the present system in biologically-relevant systems, but it is not clear from the text why DMSO is needed. The reasons given include providing "sufficient" dielectric permeability to enable control of superparamagnetic particles with a magnetic field gradient (line 191) and matching the refractive index of the silica particles (line 414). The authors may want to provide further discussion that justifies the need for greater dielectric permeability, such as qualifying "sufficient", and the need for refractive index matching between silica and solvent.
3. After stating that DMSO is used for magnetic permeability reasons, the paper next describes the

collection, transport, and delivery of E. coli by, presumably, the same superparamagnetic particles in. The methods section, but not the discussion of the results section, indicates these experiments are done in water. The authors should state clearly in the discussion that these experiments with E. coli were done in water, and why the magnetic permeability was "sufficient" here, but not for the earlier experiments.

4. The authors may want to consider citing the most recent or relevant prior art in dielectrophoretic assembly and magnetic colloid manipulation in the introduction to provide the state-of-the-art context to researchers not familiar with this area of research.

Reviewer: 1

*Demirors et al. have placed dielectric microparticles into DMSO (and a water/DMSO mixture consisting of 88% DMSO), applied an AC electric field ($f = 1$ MHz) and observed that these microparticles (termed shuttles) attract other microparticles (referred to as cargo) through electrostatic interactions. By switching the electric field off, the interaction could be turned off. The authors proceed to show that if superparamagnetic particles are used as shuttles, magnetic forces can be used to manipulate shuttles. The authors demonstrate that *E. coli* bacteria can be captured and manipulated by the shuttle particles. Comsol simulations are used to model the observations qualitatively. The authors posit that the system is an analog of intracellular cargo transport.*

Overall, the work is cute but I fail to see the major advance.

General reply: We thank the Reviewer for thoroughly reading our paper and we are glad that he/she found our work cute. The reviewer raises concern about the applicability of the method in biological systems and possible limitations, such as autonomy and the ambiguity about the potential creation of chemical gradients. In short, we would like to emphasize that our work provides a new platform for the manipulation of particles and chemicals in a controlled environment. We believe this is very attractive for the development of bio-analytical chips for drug discovery and for the study of fundamental cell-material interactions. This is actually the theme of a larger project in our group, which will follow up on the work presented in this manuscript. We address these and the other remarks of the reviewer below.

Apparently the system does not work in pure water or buffer, which makes it almost useless for biological systems.

Reply: Based on this feedback, we realized that we have not made it clear enough in the original submitted manuscript that our colloidal shuttles work perfectly in deionized water. We demonstrate in Figure 4 the transport of *E. coli* bacteria in deionized water. In response to this comment we also performed new experiments to show the possibility to transport model silica colloids in pure water, see new Supplementary Figure S4. This shows that the proposed platform works in a variety of solutions including water suitable for biological applications. This point is now emphasized in the revised manuscript with the following text on page 10:

“Because dielectrophoretic forces will apply for any volume with a dielectric contrast to the medium, our colloidal shuttle can also trap and carry bio-objects like living cells or bacteria suspended in water. Water in such a scenario will provide the necessary environment for living objects to survive but also the necessary dielectric contrast for dielectrophoretic trapping.”

The creation of field gradients of ~ 10 V/mm is difficult outside of the flat cells used by the authors. The interaction between shuttle and cargo is non-specific in a biological sense, the shuttle simply collects any microscale particle in its environment.

The principles involved in the electrostatic interaction have been worked out in detail previously by Bevan et al.

Reply: We agree with the reviewer that our method requires two electrodes, although those do not necessarily have to be flat plates. For some applications, the necessity of two electrodes may actually be limiting and their fabrication may not be an easy task. However, we still think that the advances we demonstrated in our manuscript are highly unprecedented and can directly be applied in microfluidic setups and in “organ-on-a-chip” devices, as mentioned in the main text. We believe that the here presented tools can have crucial roles to test and diagnose delivery in such micro-devices, which are otherwise very difficult. Moreover, the distance between the electrodes can easily be increased to few hundreds of microns to accommodate larger cells and even tissues, while having reasonable voltages (~20 V) and the required trapping forces. We have added new simulations in Supplementary Figure S3 for 100 μm electrode distance with 10V applied field, showing that such electrode gaps are feasible to accommodate large cells and exhibit necessary field strength.

On the non-specificity of interaction, we agree with the reviewer that dielectrophoresis, the principle we use to load and unload cargos, does not provide specific interactions between the shuttle and the cargo. However, this is precisely one of the major features of our platform, since it provides a physical mechanism for size-selective entrapment that is complementary to well-known biomolecular specificity. So, if our shuttle and cargo particles are decorated with biological molecules on their surfaces, we should be able to combine the size selectivity provided by dielectrophoresis with the chemical specificity achieved by the biomolecule. Cargo-colloids functionalized, for instance, with streptavidin and biotin molecules (commercially available today) can easily lead to such bio-specificity. However, this is beyond the focus of the current work. To emphasize this point, we have added the following paragraph to our revised manuscript on page 12:

“In addition to size selectivity, bio-specificity and chemical selectivity may also be achieved by using colloidal cargo functionalized with, for instance, streptavidin and biotin molecules.”

It is also true that dielectrophoresis (DEP) is a well-established technique and Bevan and many others used the technique. To ensure that this previous work is fully recognized, we added some recent DEP work including Bevan *et al.* However, the use of this technique to load, transport and unload colloidal cargo was thus far unexplored.

The controllability of the shuttle motion is good in free space, but an actual target will in itself affect the electric field distribution and interfere with loading/unloading.

Reply: Thanks for this interesting observation, which we think is very relevant. However, as long as the actual cargo is polarized (meaning that it changes distribution of the field around itself) there will be an attraction between the two. This interaction will depend on the size ratio of the shuttle and the cargo but also the sign of their polarization (if they had a negative or a positive polarization). In all cases as long as they are polarized the dipole of the shuttle and the cargo will always attract each other however the location the attraction (the angle theta between the dipoles) may vary. We provide now the two particle and ternary particles dipolar interaction in the new Supplementary Figure S8 together with explanation of dipolar interactions. Therefore, we believe that the DEP forces around the shuttle will always suffice

to load the cargo provided that the cargo also polarizes with the applied electric field. We now modified the Supplementary Figure S2 and added a table for the combination of different assemblies for different types of cargo and shuttle polarizations but also added the following lines to the main text on page 4:

For the colloidal shuttle with lower dielectric constant ($\epsilon < \epsilon_m$, negative polarization) used in our calculations and experiments, cargo particles with $\epsilon_p < \epsilon_m$ will be trapped at the top and bottom poles while particles with $\epsilon_p > \epsilon_m$ will be trapped at the equator, as shown in Supplementary Figs. S1 and S2. Reversed configurations should form if a colloidal shuttle with dielectric constant higher than that of the medium is used ($\epsilon > \epsilon_m$). Irrespective of the polarization of the shuttle and the cargo, there will always be an attraction between them provided that they are polarized with respect to the continuous medium in which they are suspended. However, the position of attraction will vary between the poles and the equator, depending on the combination of polarizations of cargos and shuttles.

Supplementary Figures S2 | Electric polarization induced by a colloidal shuttle under a uniform electric field and combinations of shuttle-cargo assemblies for different polarizations. (a) The colloidal shuttle polarizes the surroundings to form an electric dipole. A negative dielectric contrast ($\epsilon_p < \epsilon_m$) is assumed in this calculation. This leads to a negative polarization, which is shown as an arrow pointing down in (d). Low field regions are generated on the top and bottom while high field regions arise on the sides. This polarization effect will not differ much if the particle is situated close to the electrode surface (Fig. 1, main text). (b, c) The gradient in electric field around the colloidal shuttle spans over a length scale that is on the order of the colloid diameter, which was 8 μm in the calculations. Field strength values are shown in V m^{-1} . (d) Combinations of possible shuttle and cargo polarizations and types of assemblies depending on their polarization. This shows that provided they are polarized the cargo and shuttle will always interact but the position of the assembly will vary.

Finally, the system is lacking major aspects of biological active transport, such as its selectivity and autonomy. These aspects have been reproduced already in other colloidal systems.

Reply: We do agree that our system paper lacks autonomy. However, this is not a drawback if it is utilized for bio-analytical purposes. On the contrary, the ability to manipulate the shuttle and cargo with external controls offer possibilities that are complementary to the selectivity and autonomous aspects already reproduced in other colloidal systems. Please note also that our system also offers tunable size selectivity. We have shown such size-selectivity in Figure 3, where we can selectively transport one type of cargo or two types of cargo in a pool of large and small cargos. The selectivity is achieved by tuning the field strength, which adjusts effectively the trapping forces for the two different-sized cargos. This is a very useful control parameter for the deliberate manipulation of the cargo colloids.

The authors frequent invocation of chemical gradients seems inappropriate, since the transport of a few colloidal particles does not meaningfully change chemical potentials. Even for biological active transport, describing it in chemical terms is largely unhelpful.

Reply: We understand this concern. Our original text has not made it clear enough that the chemical gradient would not be generated by the colloidal particles themselves, but from molecules that these colloidal particles would locally release at specific sites. In the revised text we refer now to the potential to obtain chemical gradients if for example hydrogel cargos soaked with chemicals or other trigger-releasing agents in colloidal size. In the future we aim to synthesize mesoporous colloids or photo-responsive colloidal hydrogels as cargo and use them as agents to induce chemical gradients or deliver chemicals. We clarified this by adding the following lines to the text on page 15;

Chemical gradients can be created by locally releasing high concentrations of chemical species that are previously incorporated in porous or hydrogel cargo particles. The controlled release of these species from the cargo should be possible by following established strategies involving for example light or local heating as triggers. Utilizing such colloidal shuttles to controllably interact with tissues and organs recreated on a chip is another enticing possible application of this new technology.

Minor:

- 0.7 picoNewtons should be abbreviated as 0.7 pN not 0.7 pNs.
- Fig. S1: What are the y-axes labels and units on the left and middle figure.
- Fig. S2a, 1c: What are the units of the color scale?
- What is the particle size when calculating $F(DEP)$?

Reply: We thank the reviewer for spotting this. We have now corrected these missing pieces and mistakes. On the $F(DEP)$ we originally considered and mentioned 2 and 5 μm particles for the exerted DEP forces and we now provide much more details and theory on the DEP forces given in Supplementary Figure S3, S8 and S9. Here, in the new data we use 1 μm colloids for the plots and simulations.

Reviewer: 2

A key factor in enabling construction, control, and characterization of systems in the nano and microscales is the ability to perform spatial manipulation of objects in these length scales. In recent years, nano and micro-shuttles, swimmers and rockets have been developed and introduced new capabilities, including controlled directional movement and transport of molecular and nanoscale cargo.

One of the major challenges in utilizing such systems for transport is obtaining controlled mechanism for pick up and release of cargo. In this manuscript, Demirörs et. al present the use of dielectrophoresis (DEP) for controlled loading and unloading of microscale cargo onto colloidal magnetic shuttles, and the ability to load the cargo, transport the shuttle, and then unload it by combining external electric and magnetic fields.

In order to load cargo particles on the shuttle the authors utilize DEP forces between the particles that are formed when the particles are subjected to a (uniform) external electric field. The authors demonstrate using a finite element model, that when a colloidal particle (the shuttle) is immersed in a solvent with higher dielectric constant and subjected to a (uniform) external electric field, it perturbs the electric field in its vicinity. The electric field around the shuttle is reduced in the direction of the external field, and increased in perpendicular to it. The perturbation leads to a large gradient in the field around the particle, and consequently to a net DEP force, which can trap other (smaller) colloidal particles around the shuttle particle. Furthermore, the field depends on the geometry of the particles, and in anisotropic particles on the position of the particles with respect to the field.

The authors exhibit the applicability of DEP for cargo trapping using several systems of different geometries. Using time-lapse micrographs, the authors show that elongated rod-like shuttles act as 1d traps, which allows smaller spherical colloids or other elongated rods to move along the elongated shuttle. For large flat shuttles, the authors demonstrate accumulation of large number of spherical dots upon applying voltage, and a reversible transition of the spherical dots from disordered (liquid) to ordered structure (2d crystal) when the voltage is further increased. The authors also show that tuning the applied voltage can be used to form size-selective attachment of cargo particles on to the shuttle.

Combining the DEP loading-unloading mechanism with superparamagnetic shuttles allows controlled transport of the cargo against the concentration gradient. The authors demonstrate that using iron-oxide loaded polystyrene (with diameter of 4.6 micron) as shuttles allows transport of cargo (1 micron particles) to distances of tens of microns with velocities of ~1 micron per second. The authors demonstrate transport of inanimate cargo, as well as transport of non-spherical and cargo, including E-coli bacteria.

The method presented by the author offers an impressive demonstration of the potential of DEP forces for exciting applications including self-assembly, constraint diffusion, and controlled transport in the microscale under non-equilibrium conditions using external fields. The authors show the generality of the method by examining several systems of different geometries, demonstrate interesting features including 1d traps and liquid-to-crystalline transitions, and analyze the boundaries of the transport with respect to drag forces.

However, while the experimental results are indeed impressive, the treatment of the forces analysis, both theoretically and the experimentally should be further developed in order to better understand the behavior and the characteristics of the method. There are several issues which should be elaborated in the paper:

General reply: We thank the Reviewer for his/her extensive and constructive comments on our manuscript and referring our work as “impressive”. We are glad also that he/she found

our work publishable. The Reviewer raises concern about depth of the theory provided and the lack of analysis on the limits of the method, which we address below.

1. The major issue that is missing is a more comprehensive theoretical treatment of the forces. In particular, in all the calculations the authors model the electric field around the shuttle that results from the external electric field. However, they completely neglect (and ignore) the electric field around of cargo and the forces it exerts on the shuttle. Because the difference in size of the two is usually not very large, one might expect that both the cargo and the shuttle will have a mutual DEP force, which is not negligible. While exact treatment of the dynamics is complex as it includes both hydrodynamic and electrodynamic terms, there are several works that tackled the problem (for example Ai and Qian, *Journal of Colloid and Interface Science* 346, 2010, Pages 448-454), and might give better description of the systems). Such papers were not mentioned in the references, and I urge the authors to look deeper into inter-particle DEP interactions and incorporate it in this paper.

Reply: Following the suggestion of the reviewer, we now provide a more thorough description of the dielectrophoretic forces as given now in Supplementary Figures S3, S8 and S9. We believe that these changes now cover the concerns raised by the reviewer. We address the multi-particle simulations in the updated Supplementary Figures, and in the response to comment -4- below. We added the following supplementary text and the Supplementary Figures S8 to the revised version. As mentioned in the experimental section, our calculations are simplistic and do not cover hydrodynamic interactions. With the new simulations shown in Figure S8 we show that this simplification is only valid when the size ratio between shuttle and cargo is large. Finally, we included the mentioned reference (Ref. 30) in our discussion highlighted in the main text.

Supplementary Figures S8 | Finite Element Analyses of the electric field around the shuttle before and after the formation of the shuttle-cargo assembly. A shuttle-cargo size ratio similar to the experiments was used in these simulations. To aid visualization of the interactions between many polarized particles, we indicate with an arrow the dipoles induced on the colloids due to the applied electric field. Assuming a negative dielectric contrast ($\epsilon_p < \epsilon_m$) for the shuttle and cargo colloids, the

particles will become negatively polarized, as indicated by an arrow pointing down. (a) Sketch of colloids interacting via dipole-dipole interactions. (b) Summary of how two dipolar particles will behave depending on the angle θ between them. (c) First circle of cargos ($1\mu\text{m}$) before and after (right) approaching the shuttle, indicating minimum distortion of the electrical dipoles when the cargo is in close proximity to the shuttle particle. (d) Second circle of cargos ($1\mu\text{m}$) before and after approaching a shuttle that had already been surrounded by larger ($2\mu\text{m}$) cargo particles. This situation represents the configuration observed in the experiment described in Figure 3e,f (main text). The simulations reveal strong distortions of the electric field around particles if multiple particles are close to one another. Note here that in the assembly of a second circle of cargos, the interactions between the multiple particles involved become more complex, which likely affects the pick-up of these new cargos. While the interactions between the cargo of the 2nd and 1st circles are attractive, the shuttle and 2nd circle of cargo may interact repulsively depending on the angle between them. The magnitude of these interactions also change with the sizes of the shuttle and cargos. In addition, such cargo-cargo interactions are also possible when both cargo colloids are $1\mu\text{m}$ sized; however, we did not observe them at the typical field strengths ($E=0.01-0.1\text{V}\mu\text{m}^{-1}$) used in this work. The dipole moment (\mathbf{p}) induced by the local electric field (\mathbf{E}_{loc}) scales with $\sim a^3$: $\mathbf{p} = 4\pi\alpha\epsilon_m a^3 \mathbf{E}_{loc}$, where a is the radius of the particles, $\alpha = \frac{\epsilon_p - \epsilon_m}{\epsilon_p + 2\epsilon_m}$, ϵ_p is the dielectric constant of the particle and ϵ_m is the dielectric constant of the medium. Therefore, cargo-cargo assemblies are more likely to occur for larger cargos and/or at higher field strengths.

Analysis of dielectrophoretic forces between particles:

Estimations of the dielectrophoretic forces between shuttle and cargo colloids shown in the main text neglect the influence of the cargo colloids to the electrical field gradients. Here, we conduct a more thorough analysis of the dielectrophoretic forces to show that this simplification is valid when the size ratio between shuttle and cargo is large (Supplementary Figure S8). Cargo particles alter the electric field around themselves and thus develop an electric dipole similar to the one of the colloidal shuttle. To investigate the effect of this electric dipole on the interactions between cargo and shuttle, we performed additional simulations. The results displayed in the Supplementary Figure S8 show the electric field strength around the colloidal shuttle and the cargos when they are apart and after they come closer. The simulations reveal that the electric field alteration around the small cargo colloids changes indeed the field around the larger one but this effect is found to be minimal for the size ratio of the spherical shuttle and cargo particles used in our work. This justifies the simpler calculations assumed in the main text. By contrast, polarization of the cargo may play an important role when more than one circle of cargos is trapped by the shuttle, as is the case in the size-selective experiments shown in Figure 3e,f (main text). In this more complex configuration, the DEP forces around the colloidal shuttle are significantly altered after the first circle of cargos are attracted and trapped by the shuttle. This lowers the attraction for a possible second circle of cargos, which we experimentally observed at high external fields (Figure 3e,f, main text).

Besides the simulations shown in the Supplementary Figure S8, a qualitative analysis of the dipolar interactions between cargo and shuttle colloids is also possible using a simple analytical model that describes the interaction potential between dipole moments. Such

dipolar interactions are angle dependent. Attraction between the dipole of the colloidal shuttle and that of the cargo is only possible at the top and the bottom of the colloidal shuttle, which agrees with our description based on DEP forces. When two particles are electrically polarized and exhibit a dipole they interact through dipolar interactions given by

$$u_{dip}(r_{ij}) = 4kT\gamma \left(\frac{a}{r_{ij}}\right)^3 (1 - 3\cos^2\theta_{ij}),$$

where a is the radius of the particles, \mathbf{r}_{ij} is the vector between particles i and j , and θ_{ij} is the angle between the vector \mathbf{r}_{ij} and the electric field direction, k is the Boltzmann constant and T is the temperature. $\gamma = \frac{p^2}{16\pi\epsilon_m a^3 kT}$, where $\mathbf{p} = 4\pi\alpha\epsilon_m a^3 \mathbf{E}_{loc}$ is the dipole moment induced by the local electric field $\mathbf{E}_{loc} = \mathbf{E} + \mathbf{E}_{dip}$. Here, \mathbf{E} is the external electric field and \mathbf{E}_{dip} is the field induced by other dipoles. The term $(1 - 3\cos^2\theta_{ij})$ in the equation above quantifies the angle dependence of the interaction between two dipoles. When two dipoles are arranged such that $\theta_{ij} > \sim 54.74$ then these dipoles repel each other. If the angle θ_{ij} is below 54.74 then the two dipoles attract each other, making a head-to-toe configuration more favorable. Supplementary Figure S8b displays simplistic sketches of these attractive and repulsive interactions.

2. In the discussion, the authors mention that cargo attached using DEP might detach upon transport because of the drag force, and predict the maximum velocity that can be obtained before detachment. While in the scope of the paper this is an unwanted result, the detachment velocity can act as direct measurement for obtaining the DEP forces experimentally. In addition to the information that such experimental measurement will contribute to this paper, it will be an excellent tool for assessing theoretical results regarding the inter-particle DEP, which is still a subject of research. Thus, I would recommend measuring the detachment as function of velocity, and extracting an estimate for the forces involved in the DEP shuttle-cargo systems.

Reply: We thank the reviewer for these constructive and useful suggestions, which help to improve the quality and clarity of our paper. Following this suggestion, we performed additional experiments and extracted data from previous measurements to obtain the experimental values of DEP forces from the speed of the shuttle at the moment it loses the cargo as a function of electric field strength. We also performed the analytical calculations for the DEP forces (F_{DEP}) and Drag forces (F_{Drag}) and compared them with the experimentally obtained values. These are shown in the new plots given in Supplementary Figure S9. The DEP forces exerted by colloidal shuttles that successfully hold the load are always higher than the drag forces estimated for the size and experimental speed of these colloids. Colloidal shuttle starts losing cargos when the F_{DEP} is close to the F_{Drag} as demonstrated in Figure S8d and S8e. The theoretical predictions perfectly match with the experimental observations and data.

Supplementary Figures S9 | Theoretical and experimental analyses of dielectrophoretic (DEP) trapping forces (F_{DEP}) and the drag forces exert on the colloids due to their motion in liquid medium. DEP forces acting on a $1\mu\text{m}$ sized silica cargo were estimated by extracting from the finite element analysis the maximal gradient of the electric field square (∇E^2) value around the colloidal shuttle as a function of applied electric field between the electrodes. This ∇E^2 value was used to calculate the DEP forces using the relation: $F_{\text{DEP}} = 2\pi a^3 \epsilon_m \text{Re} \left\{ \frac{\epsilon_m^* - \epsilon_p^*}{\epsilon_m^* + 2\epsilon_p^*} \right\} \nabla |E|^2$. (a,b) The theoretical F_{DEP} is plotted against (a) the applied electric field strength and (b) the square of the applied electric field strength. The linear plot allows us to estimate the expected F_{DEP} for a given electric field applied in our experiments. To compare the F_{DEP} forces in DMSO and in water we plotted in (c) F_{DEP} estimates for both media. (d) From the Stokes drag formula given in the main text we estimated the drag forces exerted on a moving cargo with $1\mu\text{m}$ size. Experimental velocities of the colloidal shuttles obtained by image analyses were used in these calculations. Successful and unsuccessful cargo trapping experiments were compared in terms of the F_{DEP} and F_{Drag} values expected. For all successful transport cases (+ and \diamond in (d)) F_{DEP} exceeds F_{Drag} . However, loose cargo trapping (\diamond in (d)) was observed when F_{DEP} was on the same order as F_{Drag} . These experiments nicely demonstrate that our predictions for F_{DEP} and F_{Drag} reflects accurately the experimental observations. (e) Confocal microscopy images of the moving shuttles with strong or loose cargo trapping. Note that the cargo was strongly trapped in all experiments plotted with the (+) sign in (d), for which F_{DEP} values are much higher than F_{Drag} .

We also added the following to the main text on page 13:

Interestingly, we observe that when F_{DEP} is comparable to F_{drag} the cargo is only loosely trapped by the shuttle, which allows for shear-triggered cargo release³¹ and for the dynamic exchange with other cargo particles present in the medium (Supplementary Fig. S9e). The triggering shear forces can be controlled by either changing the speed of the colloidal shuttle or flowing a fluid around a shuttle that is locked in place using an external magnetic field. In addition to programmable cargo release, our colloidal shuttle system can thus also be used to experimentally measure dielectrophoretic forces between colloidal particles to assess the validity of theoretical predictions³⁰.

3. In all the calculations, the shuttle particle is on top of the electrode. What is the dependence of the shuttle field on the distance from the electrode? Can this approach be extended to 3d systems as well (for example transport in tissues or hydrogels)?

Reply: This is a relevant question. Our experiments and the analysis provided below show that the approach is fully functional in 3D systems. For a spacing of 100 μm the field gradients on the bottom and the top of the colloidal shuttle remain largely unchanged throughout the entire volume between electrodes. The magnitude of the field gradient and of the trapping forces stays essentially the same. For hydrogels or tissues the dielectric constant and the viscosity of the medium will be the major changes. As both systems are rich in water, we expect that the dielectric constant of the medium to be very close to that used in our experiments. However, the drag forces in such systems will be altered as a linear function of the viscosity. This change in viscosity can easily be compensated by increasing the applied field strength. Because F_{DEP} increases with $F_{\text{DEP}} \sim a^3$ (a is radius) whereas F_{Drag} increases with $F_{\text{Drag}} \sim a$, a small increase in field strength will be sufficient to overcome the viscosity effect for applications in viscous media. To cover this point in our paper, we added the following Supplementary Figure S3 to the revised manuscript:

Supplementary Figures S3 | Variation of the electric polarization of a colloidal shuttle with the elevation of the shuttle from the electrode surface. (a) From the finite element analysis, we estimated the maximal field gradient around the shuttle by measuring the change in electric field strength along the line shown in the inset. Here, we plotted the maximal gradient of the electric field square ($\nabla|\mathbf{E}|^2$) value around the colloidal shuttle as a function of electric field applied between the electrodes. This $\nabla|\mathbf{E}|^2$ value is a direct measure of the F_{DEP} ($F_{\text{DEP}} = 2\pi a^3 \epsilon_m \text{Re} \left\{ \frac{\epsilon_m^* - \epsilon_p^*}{\epsilon_m^* + 2\epsilon_p^*} \right\} \nabla|\mathbf{E}|^2$). (b) Analyses of the field gradients around the colloidal shuttle were performed as a function of distance

from the electrode until 50 μm , which corresponds to the middle between the two electrode surfaces. Except for a stronger polarization in the bottom side of the shuttle at distances very close to the electrode surface, the estimated maximal ∇E^2 stays essentially constant along different elevations. This means that the F_{DEP} forces will stay constant within the sample cell, proving that our technique operates in 3D with the same efficiency as on the surface of the electrode. A negative dielectric contrast ($\epsilon_p < \epsilon_m$, negative polarization) is assumed in these calculations.

We also added the following sentence to the main text to highlight this additional feature of the platform on page 4:

Because the electric field gradient and the DEP forces around the colloidal shuttle stays essentially constant as a function of its elevation from the bottom electrode, we are confident that this technique operates in 3D with the same efficiency as on the surface of the electrode (see Supplementary Fig. S3).

4. Figure 3, panels e and f – please provide a more detailed information regarding the image. In particular, why would the small particles (depicted in blue) will be positioned on top of the green particles? Why do the small particles seem darker in the micrograph images? Thus, I recommend this paper for publication after major corrections, I particular regarding the theoretical calculation and experimental measurements of the forces involved in the process.

Reply: This is again another good point about our system, which we clarified partially in our response to point -1- above. First of all, small particles sitting on the first circle of cargos is lying at a different elevation than the focal plane of the objective. Therefore, they look darker than the others. This is purely an optical artifact. The reason for the 2nd circle of cargos to sit on top of the 1st is due to the dipolar interactions between the cargos and the colloidal shuttle, which we describe now in Supplementary Figure S8, more specifically the 2nd circle of cargos are repelled by the shuttle whereas they are attracted to the cargos of the 1st circle.

We have added the following sentence to the main text to clarify this point on page 12:

Note here that if both types of particles are trapped, the larger cargo particles will be captured first, forming a ring at the bottom of the shuttle. The polarization of the larger cargos that are first captured by the shuttle will attract and facilitate the subsequent pick-up of the smaller cargos. While the interactions between the large and small cargo particles are attractive, the shuttle and the smaller cargo may interact repulsively, depending on the angle between them³⁰ (see also Supplementary Fig. S8).

Reviewer 3.

This paper describes a clever way to collect, transport, and deliver colloidal cargos by combining dielectrophoresis and magnetic manipulation of superparamagnetic particles. The authors demonstrate reversible capture and release of colloidal cargo by using dielectrophoretic force and directed translational motion of colloidal shuttle using magnetic field gradient. Finite element calculations are performed to understand electric field strength around the colloidal shuttle. Both dielectrophoretic and magnetic manipulation of colloids are both well-known, the combination of the two into a single system is a fascinating study of controlled transport of micro cargos. It's not clear if this occurs only in DMSO/water or if it is also amenable to more biologically-relevant environments, but the manuscript is well-organized, well-presented, and should be of broad interest. After careful reading, there are a few issues that need clarification or correction:

General reply: We thank the Reviewer for carefully reading our manuscript and we are glad to learn that she/he assesses it as “clever, well-organized, well-presented, and of broad interest”. We apologize for the unclarity about the ability to use our approach in water or in biologically-relevant media. We now clarified this issue in our manuscript as highlighted in the text below. All other concerns are also addressed.

Methods:

... μm -long and 400-nm-thick purchased from Merck KGaA, Germany (White Sapphire grade) coated with a thin layer of silica for fluorescent labeling ²⁵; and (iii) superparamagnetic colloids 4.69 μm in size purchased from Microparticles GmbH (PS-MAG-RhB-S2538). For shuttling experiments, a variety of solutions could be used. Initial experiments with particle mixtures were dispersed in a mixture of water (11.6 wt%) and dimethylsulfoxide (DMSO, 88.4 wt%) to match the refractive index of silica (1.45) or in pure DMSO. However, use of DMSO is not obligatory for such experiments as water will also exhibit similar behavior. Experiments with E-coli were performed in deionized water. *Escherichia coli* bacteria was labeled with mTurquoise2 fluorescent protein.

We also added the following short sentence to the main text on page 9:

When suspended in a solution like DMSO ($\epsilon = 47$) or mixtures of DMSO and water ($\epsilon = 80.4$) or in deionized water (see Supplementary Fig. S4), these superparamagnetic particles will have sufficient dielectric permeability contrast with the medium to generate dielectrophoretic traps while also responding to magnetic field gradients.

1. The manuscript demonstrates the dielectrophoretic assembly of smaller particles onto larger ones, but there isn't any obvious reason why smaller particles shouldn't assemble with each other. Indeed, this seems to occur for rod-shaped colloids (Figure 2b), but is not observed between silica colloids, except perhaps in the colloidal crystal phase at higher field strengths (Figure 2c). From eq 1, it could be argued that the r^3 dependence would substantially reduce the interaction between smaller particles, but does not rule it out completely. The authors may want to clarify this issue.

Reply: This is a relevant question and the answer is also partially highlighted in the question itself. The reviewer addresses correctly that the assembly formation between the cargos is possible, and we observed this at larger field strengths or for larger cargos (larger than $2\mu\text{m}$) at the typical field strengths ($E \sim 0.01\text{-}0.1\text{V}/\mu\text{m}$) we describe in our paper. We now elaborate the interaction between the cargo colloids in Supplementary Figure S8, and add the following lines to mention why we do not observe cargo-cargo assemblies for $1\mu\text{m}$ cargo used in our experiments.

-Text from the new Supp. Fig. 8:

In addition, such cargo-cargo interactions are also possible when both cargo colloids are 1 μm sized; however, we did not observe them at the typical field strengths ($E=0.01\text{-}0.1\text{ V}\mu\text{m}^{-1}$) used in this work. The dipole moment (\mathbf{p}) induced by the local electric field (\mathbf{E}_{loc}) scales with $\sim a^3$: $\mathbf{p} = 4\pi\alpha\epsilon_m a^3 \mathbf{E}_{loc}$, where a is the radius of the particles, $\alpha = \frac{\epsilon_p - \epsilon_m}{\epsilon_p + 2\epsilon_m}$, ϵ_p is the dielectric constant of the particle and ϵ_m is the dielectric constant of the medium. Therefore, cargo-cargo assemblies are more likely to occur for larger cargos and/or at higher field strengths.

2. The use of DMSO reduces the utility and applicability of the present system in biologically-relevant systems, but it is not clear from the text why DMSO is needed. The reasons given include providing “sufficient” dielectric permeability to enable control of superparamagnetic particles with a magnetic field gradient (line 191) and matching the refractive index of the silica particles (line 414). The authors may want to provide further discussion that justifies the need for greater dielectric permeability, such as qualifying “sufficient”, and the need for refractive index matching between silica and solvent.

Reply: We apologize for the ambiguity about the need to use DMSO-water mixture. The reasons we use DMSO-water mixtures is the refractive index matching of the silica colloids for better fluorescent imaging. However, this purpose is mostly required for many layer imaging in confocal microscopy of fluorescent silica colloids, which is not the case in our experiments. Therefore, we do not have a crucial reasoning to use DMSO then having experimented with these silica colloids in DMSO-water mixtures historically. We clarified this in the revised manuscript.

3. After stating that DMSO is used for magnetic permeability reasons, the paper next describes the collection, transport, and delivery of E. coli by, presumably, the same superparamagnetic particles in. The methods section, but not the discussion of the results section, indicates these experiments are done in water. The authors should state clearly in the discussion that these experiments with E. coli were done in water, and why the magnetic permeability was “sufficient” here, but not for the earlier experiments.

Reply: We apologize for the confusion about the necessity of DMSO. The use of DMSO for having a magnetic permeability contrast to magnetic colloids is indeed a true statement but we totally agree with the reviewer that this statement may be confusing, since a similar contrast exists with water and many other solvents available. Therefore, we updated the discussion and methods of the manuscript with the following lines:

-Line 191 on page 9

When suspended in a solution like DMSO ($\epsilon = 47$) or mixtures of DMSO and water ($\epsilon = 80.4$) or in deionized water (see Supplementary Fig. S4), these superparamagnetic particles will have sufficient dielectric permeability contrast with the medium to generate dielectrophoretic traps while also responding to magnetic field gradients.

4. The authors may want to consider citing the most recent or relevant prior art in dielectrophoretic assembly and magnetic colloid manipulation in the introduction to provide the state-of-the-art context to researchers not familiar with this area of research

Reply: We thank the reviewer for raising this concern. We added the following references to our manuscript to better represent the recent work in dielectrophoretic assembly and magnetic colloid manipulation.

14. Timonen, J. V. I., Demirörs, A. F. & Grzybowski, B. A. Magnetofluidic Tweezing of Nonmagnetic Colloids. *Adv. Mater.* **28**, 3453–3459 (2016).
15. Driscoll, M. *et al.* Unstable fronts and motile structures formed by microrollers. *Nat. Phys.* **13**, 375–379 (2017).
16. Bonakdar, N. *et al.* Biomechanical characterization of a desminopathy in primary human myoblasts. *Biochem. Biophys. Res. Commun.* **419**, 703–707 (2012).
17. Vlaminck, I. D. & Dekker, C. Recent Advances in Magnetic Tweezers. *Annu. Rev. Biophys.* **41**, 453–472 (2012).
20. Chiang, M.-Y., Hsu, Y.-W., Hsieh, H.-Y., Chen, S.-Y. & Fan, S.-K. Constructing 3D heterogeneous hydrogels from electrically manipulated prepolymer droplets and crosslinked microgels. *Sci. Adv.* **2**, e1600964 (2016).
21. Edwards, T. D. & Bevan, M. A. Controlling Colloidal Particles with Electric Fields. *Langmuir* **30**, 10793–10803 (2014).
22. Demirörs, A. F. & Crassous, J. J. Colloidal assembly and 3D shaping by dielectrophoretic confinement. *Soft Matter* **13**, 3182–3189 (2017).
23. Demirörs, A. F., Courty, D., Libanori, R. & Studart, A. R. Periodically microstructured composite films made by electric- and magnetic-directed colloidal assembly. *Proc. Natl. Acad. Sci.* **113**, 4623–4628 (2016).
30. Ai, Y. & Qian, S. DC dielectrophoretic particle–particle interactions and their relative motions. *J. Colloid Interface Sci.* **346**, 448–454 (2010).

REVIEWERS' COMMENTS:

Reviewer #1 (Remarks to the Author):

The authors have comprehensively addressed all my prior comments.

Reviewer #3 (Remarks to the Author):

One comment, with regard to reviewer 3 point 1: on closer inspection of the Transfer of Large and Small Cargo, it appears that smaller cargo can associate under the applied field. There are many dimers visible which fall apart when the e-field is lowered. Noticeable, but not nearly as dramatic as the assembly to the larger particle which steals the show. This doesn't take away from the main thesis of the paper, but might warrant a note to that effect in the text. At any rate, the authors have addressed this reviewers reservations and improved the quality and clarity in the revised manuscript. Pleased to recommend this manuscript for publication. Great work.